# The Predominance of the Health-Promoting Patterns of Work Behavior and Experience in General Practice Teams—Results of the IMPROVE*job* Study

**DOI:** 10.3390/healthcare12030299

**Published:** 2024-01-24

**Authors:** Beatrice Thielmann, Anke Wagner, Arezoo Bozorgmehr, Esther Rind, Achim Siegel, Melina Hippler, Birgitta Weltermann, Lukas Degen, Julian Göbel, Karen Minder, Tanja Seifried-Dübon, Florian Junne, Anne Herrmann-Werner, Karl-Heinz Jöckel, Verena Schröder, Claudia Pieper, Anna-Lisa Eilerts, Andrea Wittich, Monika A. Rieger, Irina Böckelmann

**Affiliations:** 1Institute of Occupational and Social Medicine and Health Services Research, University Hospital Tübingen, Wilhelmstr. 27, 72074 Tübingen, Germany; beatrice.thielmann@med.ovgu.de (B.T.); esther.rind@med.uni-tuebingen.de (E.R.); achim.siegel@med.uni-tuebingen.de (A.S.); melina.hippler@med.uni-tuebingen.de (M.H.); monika.rieger@med.uni-tuebingen.de (M.A.R.); 2Institute of Occupational Medicine, Faculty of Medicine, Otto von Guericke University Magdeburg, Leipziger Str. 44, 39120 Magdeburg, Germany; irina.boeckelmann@med.ovgu.de; 3Institute of General Practice and Family Medicine, University Hospital Bonn, Venusberg-Campus. 1, 53127 Bonn, Germany; arezoo.bozorgmehr@ukbonn.de (A.B.); birgitta.weltermann@ukbonn.de (B.W.); lukas.degen@ukbonn.de (L.D.); julian.goebel@ukbonn.de (J.G.); karen.linden@ukbonn.de (K.M.); 4Department of Psychosomatic Medicine and Psychotherapy, University Hospital Tübingen, Osianderstr. 5, 72076 Tübingen, Germany; tanja.seifried@med.uni-tuebingen.de (T.S.-D.); florian.junne@med.ovgu.de (F.J.); 5Tübingen Institute for Medical Education (TIME), Faculty of Medicine, University of Tuebingen, 72076 Tübingen, Germany; anne.herrmann-werner@med.uni-tuebingen.de; 6Center for Clinical Trials, University Hospital Essen, University of Duisburg-Essen, Hufelandstr. 55, 45122 Essen, Germany; k-h.joeckel@uk-essen.de (K.-H.J.); verena.schroeder@uk-essen.de (V.S.); 7Institute for Medical Informatics, Biometry and Epidemiology, University Hospital Essen, University of Duisburg-Essen, Hufelandstr. 55, 45122 Essen, Germany; claudia.pieper@uk-essen.de (C.P.);; 8Occupational Health Psychology—Research and Consulting, Sternbergstr 19, 72074 Tübingen, Germany; a_wittich@web.de

**Keywords:** primary care, physicians, practice assistant, general practices, leadership, workload, workplace

## Abstract

This study aims to identify the distribution of the “Work-related behavior and experience patterns” (Arbeitsbezogenes Verhaltens-und Erlebnismuster, AVEM) in general practitioners and their teams by using baseline data of the IMPROVE*job* study. Members of 60 general practices with 84 physicians in a leadership position, 28 employed physicians, and 254 practice assistants participated in a survey in 2019 and 2020. In this analysis, we focused on AVEM variables. Age, practice years, work experience, and working time were used as control variables in the Spearman Rho correlations and analysis of variance. The majority of the participants (72.1%) revealed a health-promoting pattern (G or S). Three of eleven AVEM dimensions were above the norm for the professional group “employed physicians”. The AVEM dimensions “striving for perfection” (*p* < 0.001), “experience of success at work” (*p* < 0.001), “satisfaction with life” (*p* = 0.003), and “experience of social support” (*p* = 0.019) differed significantly between the groups’ practice owners and practice assistants, with the practice owners achieving the higher values, except for experience of social support. Practice affiliation had no effect on almost all AVEM dimensions. We found a high prevalence of AVEM health-promoting patterns in our sample. Nearly half of the participants in all professional groups showed an unambitious pattern (S). Adapted interventions for the represented AVEM patterns are possible and should be utilized for maintaining mental health among general practice teams.

## 1. Introduction

There is an increasing workload in general practitioners’ offices [1,2]. The most mentioned reasons for increasing workload for general practitioners (GPs) and practice assistants were increasing administrative paperwork related to documentation, bureaucracy, changes in regulations, and practice management [1,2], as well as higher patient expectations and behavior regarding health issues and help-seeking [2]. In addition, studies revealed reduced job satisfaction, excessive work hours, time pressure, insufficient salary, and lack of respect [3]. Some German studies have confirmed these previous results. For example, one study by Götz et al. (2010) identified income, hours of work, and mental working conditions as relevant factors for reduced job satisfaction for GPs [4]. Siegrist et al. (2010) showed that primary care physicians in Germany reported higher work stress than the same group in the U.S. and the U.K. in an international comparison [5]. Further studies among practice assistants in Germany also revealed requests for improvement regarding income [6,7] and recognition for their work [6]. Viehmann et al. (2017) described a high prevalence of perceived chronic stress in general practitioners and practice assistants [8]. A qualitative study by Vu-Eickmann et al. (2017) indicated the following work-related stressors for practice assistants: high workload, unforeseeable incidents, relationship problems with supervisors, and lack of perceived support from colleagues [9]. 

GPs work in small private practices with high patient loads, which are characteristic in Germany [10]. A new trend regarding ambulatory care shows an increasing number of physicians not working as self-employed in their own practice but being employed in medical care centers (in German Medizinisches Versorgungszentrum (MVZ)) [11]. With a percentage of 12%, almost every eighth healthcare worker in Germany was at least 60 years old, and significantly more than a third (41%) were 50 years and older in the year 2018. The proportion of over 59-year-olds was particularly high among physicians working in practices (30.7%) [12]. Thus, not only are the patients getting older, but so are the GPs and employees in the doctors’ offices. The shortage of skilled professionals in medical practices in Germany is precarious and includes both GPs and practice assistants [13,14]. Given the very important role of primary care physicians in health care delivery in Germany, the remaining general practitioners and their teams are increasingly complaining of excessive workload.

In addition, the growing complexity of the workplace, changes in working conditions, and also changing demands placed on employees, for example, in terms of flexibility and work intensity, can be experienced as excessive demands by some of the owners and employees in a general practice [15].

How people face stressful situations can be a crucial factor for mental health. Work-related behavior and experience patterns (in German Arbeitsbezogenes Verhaltens-und Erlebensmuster, AVEM) identify specific personality traits in the way people deal with various work demands [16]. Different personality traits mean different patterns of work-related behavior and experience, which are “key indicators of the levels of emotional health” [17]. Antonovsky’s concept of salutogenesis is the resource-oriented approach of AVEM [18,19]. This concept focuses on the conditions that promote and maintain health through the identification of resources [17]. Regarding interventions, this approach allows for a more targeted use of opportunities for changing behaviors that are hazardous to health into health-promoting ones [17]. Various concepts explain the development of stress and coping strategies [20,21]. The concept of coherence experience by Antonovsky [20] and the transactional stress and coping concept of Lazarus [21] are further approaches to AVEM. For the former, the subjective feeling of being able to solve problems plays an important role in mental health. It also suggests that if you have an optimistic attitude toward life, you can develop better coping strategies [20]. The transactional stress and coping concept of Lazarus focuses on evaluations of work-related demands and how the situation is perceived. Work-related stress occurs when the demands are not in line with the available capacities [21]. Both concepts are important for explaining the relationship between work and health.

In Germany, GPs and practice assistants in general practices reported high levels of perceived chronic stress twice as frequently as the general population [8]. There is a need for stress reduction strategies for practice owners, employed physicians, and practice assistants. Although the study population is at high risk for intense exposure to work-related stress, the AVEM—as a well-established tool to assess work-related behavior and experience patterns—has rarely been applied to study German physicians [22,23,24,25] and, more specifically, in this population of practice owners, employed physicians and practice assistants.

Therefore, the aim of this research was to identify work-related behavior and experience patterns as a basis for further targeted intervention activities. Those interventions should be designed to support the change of behaviors that are hazardous to health into health-promoting ones, in addition to structural preventive measures in the primary care setting. We hypothesized that there would be significant differences between professional groups since the practice owners, in particular, bear greater responsibility, especially the economic responsibility, which in a previous study in Germany was described to be associated with additional stress [26].

## 2. Materials and Methods

### 2.1. Study Design

Our analysis is based on the work of the transdisciplinary research consortium IMPROVE*job*. In this research network, researchers from medical, social, and economic disciplines addressed questions concerning work-related stress and job satisfaction in general practice teams, using the primary care setting as an example for small enterprises. The research was undertaken between 2017 and 2021 and comprised four subprojects: (1) analysis of working conditions in primary care practices [27,28]; (2) development of the multimodal participatory intervention and feasibility study; (3) evaluation of the effectiveness of the intervention (IMPROVE*job* study) (study protocol: Weltermann et al. (2020)) [29,30,31,32]; and (4) assessment of whether and how results can be transferred to other work environments [33]. The IMPROVE*job* study was designed as a cluster-randomized controlled trial (cRCT) in German general practices (registration number: DRKS00012677) with the overall aim of improving job satisfaction among GPs and practice assistants by means of the multimodal participatory IMPROVE*job* intervention [29]. Ethical approval for the cRCT was received from the Ethics Committee of the Medical Faculty of the University of Bonn (Reference number: 057/19). In addition, the Ethics Committees of the Medical Association Nordrhein (Lfd-Nr.:2019107) and of the Medical Faculty, University Hospital of Tübingen (Project-No.: 446/2019BO2), agreed. Participants of the general practices received written information in advance and signed informed consent forms. Confidentiality and the anonymous processing of the data were guaranteed. The IMPROVE*job* invention was developed using a participative approach and included knowledge from various experts in the fields of general practice and family medicine, occupational medicine, psychosomatic medicine, medical didactics, operations research, and workplace health promotion. The IMPROVE*job* study comprised a baseline survey, the implementation of the developed participative multimodal intervention during a nine-month implementation period, and a follow-up survey. In the present study, we focus on data from the baseline survey.

### 2.2. Data Collection

For the baseline survey of the IMPROVE*job* study, data from 60 general practices with 84 physicians in a leadership position, 28 employed physicians, and 254 practice assistants were available. Participating general practices were from the North Rhine region of Germany. They were recruited from two university teaching practice networks and from non-teaching practices. The recruitment was conducted by the Institute for General Practice and Family Medicine, University of Bonn. Practices received invitations via letter, e-mail, or fax, which included the study information and a practice consent form [30]. A total of 1141 general practices were contacted, and 60 general practices agreed to participate, resulting in a response rate of 5.3% [30]. Most reasons for non-participation were no interest, no time, and no need, as a non-responder analysis revealed [30]. More information on the recruitment is described in Degen et al. (2021) [30]. The following inclusion criteria were applied: (1) registration of the owner as general practitioner of the Association of Statutory Health Insurance Physicians of North Rhine with or without affiliation as teaching practice of the University of Bonn or the University of Cologne and (2) the consent of the practice owner and at least one practice assistant to participate in the study [30]. The following exclusion criteria were defined: upcoming relocation of the practice or retirement of the owner [30]. Furthermore, practices that had participated in sub-project 2 (development of the IMPROVE*job* intervention and feasibility study of the intervention) were excluded [30]. The participants were offered an honorarium of EUR 50.00 for their time and effort in case they participated in the complete cRCT [29]. The data collection for the baseline survey was completed in January 2020 and thus took place before the COVID-19 pandemic started in Germany [29,30].

### 2.3. Measurements

The baseline survey of the IMPROVE*job* study covered detailed measurements regarding the following topics: job satisfaction, working conditions, leadership, general health, work-related behavior and experience patterns (AVEM), occupational safety climate, perceived chronic stress, stress coping strategies, work organizational issues, as well as team activities and roles [29]. Several baseline data on job satisfaction, perceived chronic stress, and work–privacy conflict have already been published elsewhere [30,31] without reporting the AVEM results.

The following socio-demographic data were collected: age (in years), gender (male/female/diverse), professional group (practice owner/employed physician/practice assistant), work experience, practice years, and working time (full-time/part-time) [31].

To answer our research aim, we refer to the short version of the work-related behavior and experience patterns (AVEM) by Schaarschmidt and Fischer [16,34] and socio-demographic data.

### 2.4. Work-Related Behavior and Experience Patterns (AVEM) by Schaarschmidt and Fischer

The short version of 44-item AVEM identifies eleven work-related behavioral and experience dimensions from the following three areas [17]:Engagement with work (dimensions: subjective importance of work, work-related ambition, willingness to work until exhausted, striving for perfection, and distancing ability);Resilience in dealing with the everyday stress of work (dimensions: distancing ability, tendency to resign in the face of failure, proactive problem-solving, and inner calm and balance);Emotions associated with work and life in general (dimensions: experience of success at work, satisfaction with life, and experience of social support).

Distancing ability is an important component of both the first (negatively correlated with engagement) and second areas (positively correlated with resilience) [17].

High values indicate a high level of expression for each dimension. Stanine scores (a 9-point standard scale) are used for the evaluation. Regarding the AVEM, the normal Stanine score ranges from 4 to 6 points.

According to Schaarschmidt and Fischer [17], the subjects were classified into risk patterns (A, B) and health-promoting patterns (G, S) based on the levels of expression for all dimensions. The expressions of the patterns were defined as follows:“full” expression (a pattern of >95% expression);“accentuated” expression (a pattern between >80% and ≤95% expression);“tendential” pattern expression (a pattern between >50 and ≤80%, not including a second pattern with >30% expression).

The individual AVEM patterns differ in the eleven dimensions. The typical characterizations of the four AVEM patterns are as follows [17]:Overexertion risk pattern A: excessive work engagement, limited distancing ability from work-related problems and enjoyment of life, and reduced emotional resilience to stress. → Intervention required from a health perspective;Burnout risk pattern B: reduced work engagement and emotional resilience to stress, limited distancing ability from work-related problems and enjoyment of life, and marked tendency to resign. → Intervention required from a health perspective;Healthy pattern G: clear, but not excessive, work engagement and good distancing from work-related problems with proactive problem-solving; in summary, resilience to stress with positive feelings about life. → Intervention from a health point of view is not required;Unambitious pattern S: low level of work engagement, marked distancing ability from work-related problems and ‘taking it easy’, and emotional resilience to stress. → Intervention from point of view of motivation can be recommended.

Combination patterns with two predominant patterns, both total > 80%, with the mildest pattern > 30%, or “unclassifiable” patterns (none of the above criteria apply) are not used in this publication because the manual of Schaarschmidt and Fischer [17,35] does not list combination patterns.

The AVEM is an extensively validated method that has been tested on various samples (e.g., teachers, nursing staff, educators, etc.) [17]. There were conclusive relationships with characteristics of other questionnaire procedures, e.g., Freiburg Personality Inventory, Maslach Burnout Inventory, Stress Processing Questionnaire, etc. [17]. 

### 2.5. Statistical Analysis

Statistical analyses were conducted using SPSS Statistics 28 (IBM, New York, NY, USA). A test for normal distribution and a descriptive analysis of the total sample were performed. Cross-tables and cross-tabulations with Pearson’s chi-square tests were carried out for gender and working time. For the evaluations of the age, practice years, and AVEM dimensions (interval scale Stanine scores) between professional groups, the Kruskal–Wallis test was used first. If statistical significance existed, the post hoc Bonferroni correction was applied. The following significance level was defined: *p* < 0.05, *p* < 0.01, and *p* < 0.001. Finally, Spearman’s correlation analyses and a mixed (multilevel) model (Generalized Linear Model (GLM)) on all AVEM dimensions considering the variables practice owner, employed physicians, and practice assistants with assessment of the respective intraclass correlation as measure for effect size were performed. A mixed-effects model is a statistical model containing both fixed effects and random effects. Potential cluster effects on practice level were thereby adequately considered.

The Spearman’s correlation coefficients (rho) were interpreted according to Akoglu [36]: <0.39 means a low effect, 0.40–0.59 means a moderate effect, and >0.6 means a strong effect. In the analysis of variance, according to Cohen [37], the interpretation of η² can be 0.01 (small effect), 0.06 (medium effect), or 0.14 (large effect).

## 3. Results

### 3.1. Socio-Demographic Distribution of the Sample

The sample included 366 (N) physicians and employees from general practices. Due to the high proportion of practice assistants who are predominantly female, there is an overrepresentation of women (n = 319, 87.2%, mean age 43.0 ± 12.80 years) in our sample. The percentage of men was 12.8% (n = 47, mean age 53.6 ± 8.11 years). The mean age of the total sample was 44.4 ± 12.79 years, ranged 18–69 years (n = 365). The sample was divided into the following groups of participants: 84 (23.0%) practice owner physicians, 28 (7.6%) employed physicians, and 254 (69.3%) practice assistants. Further details on these subgroups are shown in Table 1.

### 3.2. Distribution of Subjects among the AVEM Groups

The 366 subjects (total sample) were represented as follows: 72 with a “full” expression of one pattern (n = 6 pattern A, n = 15 pattern B, n = 9 pattern G, and n = 42 pattern S);94 with an “accentuated” expression of one specific pattern (n = 5 pattern A, n = 21 pattern B, n = 23 pattern G, and n = 45 pattern S);60 with a “tendential” expression of one specific pattern (n = 8 pattern A, n = 8 pattern B, n = 20 pattern G, and n = 24 pattern S);52 with a pattern combination (n = 9 G/S, n = 7 S/G, n = 5 G/A, n = 7 A/G, n = 6 S/B, n = 11 B/S, n = 4 A/B und n = 3 B/A);88 subjects for whom no assignment was possible.

The consideration of the surveyed medical personnel in the three professional groups showed that 19.5% of the study participants had the risk pattern of AVEM B. Additionally, 8.4% of the medical personnel had the AVEM risk pattern A, whereas 72.1% of subjects (23.0% with G and 49.1% with S) showed a health-promoting AVEM pattern (for information on the distribution of AVEM patterns in the professional groups, see Table 2). 

### 3.3. Expressions of the AVEM Dimensions Considering Professional Groups

Figure 1 shows the expressions of the individual AVEM dimensions. Since the individual expressions of the AVEM dimensions are essential for classification into one of the four AVEM groups, highly significant differences are to be expected. These were, therefore, neither calculated nor presented. 

Table 3 shows an overview of the expressions of the individual AVEM dimensions of the different professional groups. Some AVEM dimensions were outside the norm according to the reference population. The norm of the Stanine scores ranges from four to six points, according to Schaarschmidt and Fischer [17]. The AVEM dimensions “Subjective importance of work” and “Proactive problem-solving” were slightly lower, and the AVEM dimension “Distancing ability” was slightly higher than the norm for the professional group employed physicians. In the professional group of the practice owners, “Striving for perfection” was slightly below the normal limit. 

Four AVEM dimensions differed significantly between the groups’ practice owners and practice assistants (“striving for perfection” (*p* < 0.001), “Experience of success at work” (*p* < 0.001), “satisfaction with life” (*p* = 0.003), and “experience of social support” (*p* = 0.019)). It turns out that employed physicians and practice assistants showed fewer “Experience of success at work” than practice owners (*p* = 0.008 and *p* < 0.001). On the other hand, the practice assistants showed a higher Stanine score for the dimension “striving for perfection”, which is to be evaluated negatively according to Schaarschmidt and Fischer [17] to a greater extent than the other two groups. 

The gender distribution within the AVEM patterns is shown in Table 4. 

### 3.4. Factors Associated with the Expression of the AVEM Dimensions

For the Spearman Rho correlation, the independent variables age, practice years, work experience, and hours per week (full-time/part-time) were considered (Table 5). There were moderate negative correlations at the 0.01 level for the dimensions “work-related ambition” and “work experience” (ρ = −0.433 **). Other low positive or negative correlations are shown in Table 5. 

The results of GLM analysis of mixed models examining the effect of practice affiliation (i.e., cluster) as a random effect and person group (professional groups) as a fixed effect revealed very low values for intraclass correlation (see Table 6). For the AVEM dimensions “striving for perfection” and “experience of success at work”, the iteration was terminated without achieving convergence. In summary, practice affiliation explained less than 5–10% of the total variance for almost all AVEM dimensions except for “distancing ability” (14.2%). Statistically significant effects of the three professional groups were found only for the dimensions “Work-related ambition” (with the lowest mean value in practice assistants) and “Distancing ability” (with the highest mean value in employed physicians) (Table 6).

## 4. Discussion

Baseline data of the IMPROVE*job* study were used to investigate work-related behavior and experience patterns of general practice teams in the North Rhine region of Germany. When interpreting the results, it must be kept in mind that the data come from the baseline survey of an intervention study and not from a pure survey study. All professional groups in our study exhibited predominantly AVEM health-promoting patterns G or predominantly S (72.1%). Risky AVEM patterns (i.e., A or B) were shown in 27.9% of the total group. Regarding our hypothesis, we found only few differences between the professional groups. This could be due to the fact that persons with similar personality traits find each other and work successfully together in a practice, just as we see similar pattern distributions across the professional groups in our study. The practice owners showed the highest values in the AVEM dimensions “experience of success at work” and “satisfaction with life”. Here, the Stanine scores of both variables were above the normal range (see Table 3) for practice owners. In contrast, the AVEM dimension “striving for perfection” revealed a higher value for practice assistants than for practice owners. The Stanine scores of the health-promoting AVEM dimension “Experience of social support” showed higher values for practice assistants and employed physicians compared to practice owners. The high values for this dimension are characteristic of the G and S patterns. These identified differences in the AVEM dimensions can be explained by the different areas of work of the professional groups. Practice assistants are probably confronted with more dissatisfied patients than practice owners [38] and, therefore, receive less recognition for their work. Thus, lower scores for “experience of success at work” and “satisfaction with life” in this group are not surprising. Furthermore, practice assistants are employees of the practice owners and, therefore, are in a more dependent role. It is thus reasonable to assume that practice assistants intend to work as “perfectly” as possible in order to satisfy their employer. They also work more in groups and can, therefore, receive more social support than practice owners.

To interpret the individual dimensions (health-promoting vs. health-risky), an individual evaluation of the significance of work in relation to other areas of life (personal area) should be carried out.

Regarding the gender distribution in our study, we have an overrepresentation of women versus men in our sample (n = 319 versus n = 47). The health-promoting pattern S was primarily represented by both women (n = 94) and men (n = 17). The men surveyed here also exhibited the patterns G, A, and B in quite equal amounts. Women in our sample also showed pattern G (n = 47) and B (n = 40); pattern A was less represented among women (n = 15). The representation of pattern B should be taken into account, as women often have a higher risk for burnout due to the multiple burdens of work, children, and care work.

Practice affiliation had less than a 5–10% effect on the results. Work experience had moderate negative correlations at the 0.01 level for the “work-related ambition” dimension of AVEM; otherwise, only weak correlations were detectable. This may indicate that work-related ambitions decrease with longer work experience. But more studies investigating this assumption are needed. Working time (hours per week) was associated with the AVEM dimensions “distancing ability”, “work-related ambition”, and “satisfaction with life”.

Half of the participants in all three professional groups showed an AVEM S pattern. Compared to other professional groups, the percentage of AVEM pattern S in our sample was higher. AVEM S was not present in the professional group of university lecturers [39] and reached 34% in policemen [23], 34% in prison officers [23], 43% physicians in hospitals [23], 25% in teachers [23], 47.1% in nurses [40], 61.4% in psychiatric nurses [40], 22% in medical students [41], and 46.2% in employees of international financial services companies [42]. Other cross-sectional studies showed a prevalence of pattern S of 37% in ambulance service personnel and 23.5% in administrative employees of a big city [43]. The average work experience of practice owners and practice assistants in our study was over 10 years compared to employed physicians. It may be an indication that more experienced staff cope better with the demands of the job and therefore probably develop a pattern S. Physicians and medical staff are confronted with various requirements in their profession: high quality, safe and patient-centered care considering an increasing body of knowledge, clinical and practical skills, and learned professional values within the job occur within their professional field and consequently increases with work experience. Decision-making is based on evidence-based medicine and takes into account the patient’s circumstances and preferences and is carried out using available resources [44].

More than a quarter of the participants (27.9%) in our study had risky AVEM patterns A or B. Compared to results in German physicians or other professional groups explored in previous studies, the results assessed in our study were lower than the proportion of AVEM risk patterns A and B previously published. For other professional groups, the proportion of AVEM risk patterns A or B were described as follows: for example, teachers 40–80% [45,46], 65% university lecturers [39], 34% policemen [23], 38% prison officers [23], 40% physicians (mainly working in hospital settings) [23], 39–43% physicians working in private practice (surveyed in 2008 and 2010) [25], 47% psychotherapy trainees [47], 41% nurses in hospital setting [40], 38–50% geriatric nurses [48,49], 69% medical students [41], and 34% employees of international financial services company [42]. Some of these studies have used the long form of the AVEM with 66 items. However, this difference is negligible because the intercorrelations of the corresponding scales of the standard and short forms are between 0.95 and 0.97 [34]. Thus, compared to previous studies, our sample revealed an underrepresentation of the AVEM risk patterns A and B. These comparisons look good at first glance, but Schaarschmidt and Fischer usually recommend the unambitious AVEM pattern S interventions that are based on the enhancement of motivation [17]. Taking the patterns A, B, and S together, 77% of all subjects exhibited AVEM patterns that could benefit from intervention with regard to health-related outcomes [17].

### 4.1. Discussion on the Importance of AVEM Patterns and the Need to Take Action

The AVEM pattern S is intended to indicate an unambitious attitude, which, in this case, characterizes the relationship toward work. Schaarschmidt and Fischer pointed out that this unambitious attitude was not a sign of resignation but rather the relatively high and favorable expressions in the dimensions of the area emotions toward occupational stress [17]. Overall, a positive attitude to life (relatively high life satisfaction) is described for this group [17]. However, the source of this is probably to be found outside of work [17]. In our sample, we detected mostly the pattern S. This result is matched by the comparatively low value for the dimension “Subjective importance of work” (lower normal limit). It can be assumed that the increasing bureaucracy and requirements of associations of statutory health insurance physicians in Germany (e.g., flat rate per case, prescription guidelines, etc.) lead to unambitious behavior. The value of medical healing and the importance of the work are reinforced by successful work and patient satisfaction, even though there are so many demands. This involves an evaluation of the advantages and disadvantages, taking into account the pleasant and unpleasant sides of work requirements.

The failure to meet needs and expectations in the work environment often leads to a reduction in the level of expectations, which then results in what is known as “resigned job satisfaction” [50]. It is the consequence of an unconscious lowering of individual demands regarding working conditions [51]. A qualitative analysis of job satisfaction in Germany (hospital setting) revealed that only one in four employees, and among physicians, only one in ten, was satisfied with their job [51]. However, this does not mean that those affected quit their job. On the contrary, the study by Hiemisch et al. shows a clear discrepancy between the information provided by employees and the terminations actually carried out [51]. It also showed that employees’ real job satisfaction is sometimes difficult to capture with traditional questionnaire items [51] 

The attitude behind the pattern S often revealed a protective function (e.g., protection against being overtaxed by inadequate working conditions, by too much emotional stress, or also by a stressful working climate) [17]. This is valid for all professional groups examined here. It can be assumed that no serious health disorders are to be expected with regard to the AVEM S pattern. Although, only 32% of the teachers with an AVEM pattern S indicated that “my health is good” [52]. A study showed that 8.5% of physicians with AVEM pattern S shifted to risky AVEM pattern B [25]. Interventions that emphasize their importance for the well-being of patients despite many high professional demands (e.g., regulatory requirements, increasing bureaucracy, etc.) are helpful for pattern S and could also be useful for our sample in general practices. 

The use of the AVEM patterns in our study was reasonable because impaired mental health was observed, especially in the risk patterns of AVEM. A quarter of the subjects in our study exhibited them. AVEM risk pattern was associated with a higher effort–reward ratio, over-commitment, and dissatisfaction with everyday work or worries/concerns [47]. Another study showed a negative association not only between AVEM risk pattern B and work satisfaction but also between AVEM pattern S and work satisfaction [53]. AVEM risk patterns correlate with higher stress levels, lower recovery, and more physical, psychological, and social communication impairments [54]. Regarding ambulatory health care, the results of two repeated surveys (2008 and 2010) among statutory health insurance physicians in northern Germany working in private practices (about 35% general practitioners) seem of special interest [25]. In that study, the physicians rated various aspects for or against the establishment of private practice significantly differently depending on their work-related behavior and experience patterns (*p* < 0.05). The responses regarding factors for and against establishing and working in private practices differed significantly between physicians who exhibited either pattern G or S and those who were at risk for overexertion (pattern A) or burnout (pattern B). This was revealed, e.g., regarding the significance of the factors “patient care” (5 aspects), “work organization and work development” (3 aspects), or “special stress factors” (4 aspects), such as time stress or emotional strain. Finally, “prestige of primary care” was rated significantly more often as a factor regarding establishing and working in private practice by physicians exhibiting pattern A or B than by physicians showing pattern G or S [25]. Based on this research, the AVEM patterns seem to have special significance regarding ensuring outpatient care by physicians in private practices. Among them, general practitioners especially are of outstanding importance to the healthcare system in Germany, as was particularly evident in the context of the COVID-19 pandemic [55,56,57]. AVEM patterns can change into other patterns depending on work tasks and coping, e.g., from health-promoting to AVEM risk patterns and vice versa [25]. The study also showed decreased job satisfaction among physicians over a follow-up of about two years [25]. In particular, administrative/organizational requirements are often cited as stressful. These lead to dissatisfaction with the work situation [1,2,58]. In a study of start-ups, AVEM risk pattern A was found to be relatively stable over time and on repeated measures [17,59]. The authors recommended health promotion measures because spontaneous remission was not expected. In the context of interventions, this should include an adequate balance between exhaustion and recovery, the strengthening of personal resilience through the further development of competencies, and the establishment of favorable skills and conditions for emotional and social support [17,59]. 

According to Schaarschmidt and Fischer [17], the predominant feature of risky pattern A is a high level of effort that is not accompanied by correspondingly positive feelings about life. The excessive work engagement coupled with a low level of resilience to stress and somewhat negative emotions in persons with AVEM pattern A is a risk to health because they might push themselves too hard. Risk pattern B is the most risky pattern in which feelings of being overstretched, exhaustion, and resignation are predominant. Thus, this pattern contains the symptoms commonly seen in the final stages of burnout development. The prevalence of burnout among physicians in Germany varies between 4% and 20%, according to a recent review [60]. In a current cross-sectional study, one-quarter of the participating German GPs indicated a high prevalence of work-related burnout symptoms [46], and this proportion was considerably higher than among German nurses (5%) or German medical assistants (1%) [61]. 

### 4.2. Recommendations for Health Promotion and Management

The identification of individual behaviors that are beneficial or detrimental to health offers an advantage when planning workplace health promotion and health management measures [62]. Depending on the AVEM pattern, Schaarschmidt and Fischer recommend different intervention measures [34]. This means that individual interventions can be designed. 

For the S pattern, the necessity of interventions to increase motivation can be considered and should be adapted to the respective target group. It should be noted that the S pattern can also have a protective function, for example, in professions such as caregivers or firefighters [35]. Motivating factors for all professions can, for example, include dealing with challenging work tasks, the use of functioning social relationships in the workplace, and effective support systems for coping [35]. However, it should be kept in mind that different social groups and cultures have varying norms regarding motivation and work commitment.

In addition to the two AVEM risk patterns, A and B, a targeted intervention is also recommended for the S pattern. The above-mentioned unconscious lowering of individual expectations as a form of resigned job satisfaction can result in a lack of motivation and work ethic, with corresponding losses in quality and profit for the company [63,64].

The proportion of AVEM risk patterns was lower than the health-promoting AVEM patterns, G and S, but measures should still be taken to avoid occupational withdrawal or prolonged incapacity of the members of the practice staff. Common recommended general behavioral intervention measures for AVEM risk patterns A and B are [34]: Relaxation and compensation or implementing health-promoting habits (e.g., sport, relaxation exercise, gardening…);Training/promotion of enjoyment and experience of satisfaction against restricted feelings of life or general dissatisfaction;Re-evaluation and realistic goal setting of work tasks against experience of failure or tendency to resign;Individual stress analysis and management with learning short-term and long-term coping strategies for increasing the feeling of relaxation;Promotion of team spirit and team mentality and the creation of a positive working environment with organization and care of social contacts, also in leisure time against the experience of inadequate social support;Learning to say no, changing habits of work within an organization, balancing and time management of the demand of work, and carrying out domestic duties and leisure activities against demanding too much of oneself;Balance between work and other areas of life against excessive willingness to work until exhausted;Reduce conflicts, irritations, and impatience with stress management training, such as increased tolerance of frustration for higher satisfaction;Management and training of communication (e.g., proactive, conflict-solving) for higher communication and proactive problem-solving;Individual coaching (e.g., emotions, anxiety, self-awareness, self-confidence) against resignation, hopelessness, and despair.

In our sample, the majority of our participants revealed predominantly AVEM health-promoting patterns G or predominantly S (72.1%). Risky AVEM patterns (i.e., A or B) were shown in 27.9% of the total group. Thus, a combination of the above-described behavioral prevention measures complemented by structural prevention measures is needed to adequately address the identified patterns in our sample. The importance of the combination of behavioral and structural components is also described in other prevention areas [65]. Dealing with the various AVEM patterns represents a major challenge for general practices. Health-promoting interventions in the workplace should target this issue.

In summary, more research is needed on workplace interventions that examine AVEM patterns before and after health-promoting interventions. One previous study showed that an intervention program for nurses resulted in a significant reduction in the scales of the subjective importance of work, willingness to work until exhausted, and striving for perfection, as well as a significant increase in distancing ability and satisfaction with life [66]. The use of a mind–body therapy over six weeks affected 24% of subjects to change from an AVEM risk pattern to a health-promoting pattern [67]. However, it should be noted that no long-term follow-up was performed after the intervention. The need for targeted structural and behavioral prevention is illustrated by the results of the repeated survey among German physicians in private practice [25]. Here, the majority of the physicians presenting with pattern S (41.5%) or B (22.5%) at T1 still had this pattern two years later (T2). Less stable was the healthy pattern G, where 26% changed to pattern S, as well as the risk pattern A, where 22,6% changed to the risk pattern B. Yet, some physicians also changed from A to G (19.4%) and from B to S (18.6%) [25].

In the IMPROVE*job* study, the multimodal participatory IMPROVE*job* intervention addressing structural as well as behavioral prevention was implemented after the baseline survey with the overall aim of improving job satisfaction and reducing commonly perceived psychosocial stressors in GPs’ practices [29]. Regarding behavioral prevention, the IMPROVE*job* intervention addressed several of the aspects recommended for individuals exhibiting the AVEM risk patterns A or B as well as AVEM pattern S as described above. Yet, the intervention was not designed for an individualized prevention approach but instead addressed well-known aspects (see below). Consequently, the study participants did not receive any feedback on their individual AVEM pattern nor any recommendation regarding single components of the IMPROVE*job* intervention. Instead, the general practitioners and their teams were encouraged to focus on one or two prevention goals after the two initial workshops to ensure a participatory approach.

The IMPROVE*job* intervention focused on different topics regarding leadership, teamwork, social relations at work, occupational health and safety, and workplace health promotion, as well as work organization and work process, and included the following components [29]:Two leadership workshops (workshop 1 for practice leaders only and workshop 2 for practice leaders and their teams);A toolbox with supplemental material;A nine-month implementation period supported by IMPROVE*job* facilitators.

The intervention can be classified as comprehensive since it covers behavioral as well as structural aspects of prevention in the context of workplace health management [29,30,32,33]. Structural aspects of the intervention focused on health prevention with regard to workplace design, the workplace, work equipment, and the rest of the working environment in general practices. This was covered, for example, by the improvement of work organization in practices and the implementation of relevant occupational health and safety measures [29]. The workshops within the IMPROVE*job* intervention were positively rated by the participants, but the intention to treat analysis revealed no significant improvement in the main outcome of job satisfaction [32]. Based on qualitative data derived from the formative evaluation of the IMPROVE*job* study, the authors of the research consortium assumed a major influence of the COVID-19 pandemic on the effectiveness of the IMPROVEjob intervention [32]. For example, workshops were performed as online workshops, and online material was provided instead of having practice visits by the IMPROVE*job* facilitators [32] because of the strict legal regulations that were applied in Germany during the COVID-19 pandemic in 2020 and partly in 2021.

Based on the present analysis carried out only after the evaluation of the effectiveness of the IMPROVE*job* intervention, the very high proportion of AVEM health-promoting patterns S or G and the existence of risk patterns A and B in practice owners, as well as employed physicians and practice assistants, could be an additional reason for the lack of effects revealed as a result of the cRCT. 

### 4.3. Limitations and Strengths

Due to the voluntary nature of participation in our study, we cannot exclude the possibility that general practices with a very high perceived chronic strain and stress did not participate in this study. In addition, we only achieved a moderate sample size, with 366 physicians and staff participating in this study. Due to the moderate number of participants in our sample, we believe that the external validity of the results is rather low, and the results may only be applicable to other general practices in Germany. Furthermore, we had an overrepresentation of women in our sample. Due to the smaller number of men, no generalization of our results is possible. The comparably high proportion of individuals with AVEM pattern S in our study might point to a selection bias. Practice owners or team members who had already established coping mechanisms could have especially been interested in participating in this study, which could explain the high proportion of pattern S in the study sample. Also, our results are based on self-reported data at one particular point in time, so we cannot draw any conclusions about causal relationships. For this analysis, we did not use any additional objective data to support the survey results. Thus, a response bias cannot be excluded. Unfortunately, the AVEM questionnaire was not used in the follow-up survey to keep this questionnaire as short as possible in order to foster high participation rates. Thus, no statement can be made as to whether the IMPROVE*job* intervention may have had an impact on the proportion of the identified patterns in our sample. 

Furthermore, the AVEM questionnaire was originally not tested and validated with physicians and other self-employed persons. The AVEM questionnaire has so far been used predominantly with employees so that the situation or personality of persons working voluntarily as self-employed persons, as is the case for practice owners, might not be sufficiently well represented. So far, previous studies have shown differences in personality between self-employed persons and employees. Self-employed persons try to achieve more autonomy, success, and control [68]. Therefore, they often face longer working hours, greater work–family conflict, and higher work stress than employees [68]. A recent systematic review also discovered hints of a link between self-employment and increased risk of mental illness [69]. 

The use of the AVEM patterns in our baseline questionnaire seems reasonable and is important because impaired mental health is observed especially in the risk patterns of AVEM. In our analysis, we identified that the majority of our respondents had the unambitious pattern S, which could be interpreted as a first step within the coping process toward a more health-promoting behavior or even the health-promoting pattern G. Special interventions to address this specific pattern can therefore be useful and are recommended. One-third of our participants revealed risk patterns A or B. This indicates that there is a need for special workplace interventions to deal with stress and increased workload in this sample. GPs and practice assistants in Germany reported high levels of perceived chronic stress [8,30], and GPs were also confronted with burnout and low job satisfaction, with an emphasis on employed physicians [70]. Thus, this population crucially needs targeted approaches and measures in the context of workplace health management. Our study has made a first contribution, especially regarding practice assistants, as it identified the distribution of AVEM patterns among German GPs and their teams.

## 5. Conclusions

The work-related behavior and experience patterns of German general practice teams exhibited predominantly AVEM health-promoting patterns (72.1%) but also the presence of AVEM risk patterns A and B. The prevalence of these health-threatening AVEM patterns needs special consideration. It does not matter how high the expression of the AVEM patterns is (e.g., full, accentuated, or tendential): structural prevention as well as behavioral health promotion and management measures are needed to counteract work-related stress. The AVEM can provide information, especially on the need for targeted, individualized behavioral interventions, through the patterning and expressions of the AVEM dimensions. Nonetheless, further studies, especially in general practices, are needed to assess the effectiveness and sustained effects of these intervention measures in combination with structural preventive measures and facilitation of their implementation.

## Figures and Tables

**Figure 1 healthcare-12-00299-f001:**
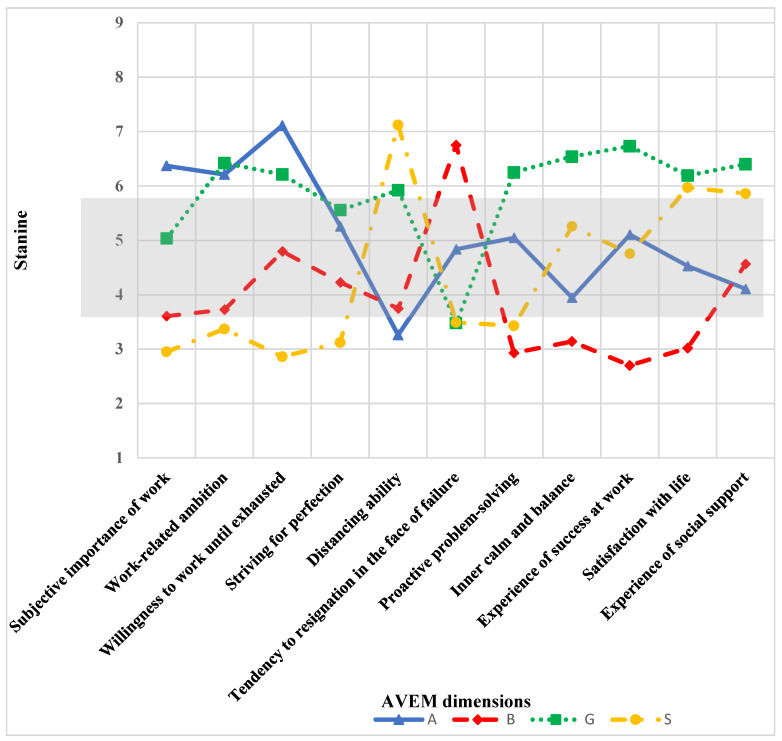
Differences in AVEM dimensions according to the 4 patterns of work-related behavior and experience (frequency of patterns in study sample: A n = 19, B n = 44, G n = 52, and S n = 111).

**Table 1 healthcare-12-00299-t001:** Socio-demographic data of the sample (presentation modified according to Table 1 in Degen et al. (2021)) [30].

	Practice Owner	Employed Physicians	Practice Assistants	Significance	
Variable	Mean ± SD Median (Min–Max)			p_Kruskal-Wallis_	Post Hoc p_Bonferroni_
Age	54.3 ± 6.18 55 (40–69)	44.8 ± 9.83 44 (30–69)	41.0 ± 12.99 42 (18–68)	<0.001	Practice owners/practice assistants < 0.001 Practice owners/employed physicians < 0.001
Practice years	16.8 ± 8.32 16 (2–36)	5.5 ± 5.47 3 (1–22)	10.4 ± 8.84 7 (1–41)	<0.001	Practice owners/practice assistants < 0.001 Practice owners /employed physicians < 0.001 Employed physicians/practice assistants = 0.007
	**n (%)**			**p_χ²_**	
n	84 (23.0)	28 (7.6)	254 (69.3)	-	
Male	40 (47.6)	6 (21.4)	1 (0.4)	<0.001	
Female	44 (52.4)	22 (78.6)	253 (99.6)		
Full-time	76 (90.5)	8 (28.6)	102 (41.5)	<0.001	
Part-time	8 (9.5)	20 (71.4)	144 (58.5)		

Notes. Abbreviations: SD = standard derivation; p_χ²_ = Pearson chi-square.

**Table 2 healthcare-12-00299-t002:** Distribution of the AVEM groups in the professional groups.

	Practice Owner/ Physicians	Employed Physicians	Practice Assistants	Total Sample
AVEM Pattern	n (%)			
A	7 (13.0)	0 (0)	12 (7.9)	19 (8.4)
B	7 (13.0)	4 (20.0)	33 (21.7)	44 (19.5)
G	13 (24.1)	6 (30.0)	33 (21.7)	52 (23.0)
S	27 (50.0)	10 (50.0)	74 (48.7)	111 (49.1)

Notes. p_χ²_ = 0.517. Pattern combinations or subjects without assignments were not considered.

**Table 3 healthcare-12-00299-t003:** Expressions of the AVEM dimensions according to professional groups.

AVEM Dimension	Practice Owner	Employed Physicians	Practice Assistants	p_Kruskal-Wallis_	Post Hoc p_Bonferroni_
	Mean ± SD Median (min–max) (95% Confidence Interval)				
Subjective importance of work	4.2 ± 1.98 4 (1–9) [3.71–4.60]	3.8 ± 2.00 4 (1–9) [2.97–4.63]	4.08 ± 2.24 4 (1–9) [3.78–4.39]	0.821	
Work-related ambition	4.9 ± 2.02 5 (1–9) [4.44–5.35]	5.0 ± 2.06 5.0 (1–8) [4.15–5.85]	4.6 ± 2.16 4 (1–9) [4.30–4.88]	0.386	
Willingness to work until exhausted	5.1 ± 2.50 5 (1–9) [4.50–5.63]	5.0 ± 1.97 5 (1–9) [4.15–5.77]	4.5 ± 2.31 5 (1–9) [4.15–4.78]	0.109	
Striving for perfection	3.6 ± 2.05 3 (1–9) [3.12–4.04]	4.4 ± 2.10 4 (1–9) [3.49–5.23]	4.7 ± 2.11 4 (1–9) [4.44–5.00]	<0.001	Practice owners/practice assistants < 0.001
Distancing ability	5.8 ± 2.23 6 (1–9) [5.25–6.26]	6.1 ± 1.66 6 (3–9) [5.40–6.76]	5.7 ± 2.20 6 (1–9) [5.39–5.98]	0.661	
Tendency to resign in the face of failure	4.3 ± 2.25 4 (1–9) [3.81–4.83]	4.9 ± 2.24 5 (1–8) [4.00–5.84]	4.5 ± 2.28 5 (1–9) [4.22–4.83]	0.097	
Proactive problem-solving	4.1 ± 2.19 4 (1–9) [3.58–4.57]	3.9 ± 2.25 4 (1–8) [2.99–4.85]	4.3 ± 2.20 4 (1–8) [4.04–4.63]	0.549	
Inner calm and balance	5.0 ± 2.03 5 (1–9) [4.52–5.43]	5.4 ± 1.75 5 (1–8) [4.64–6.08]	5.0 ± 1.97 5 (1–9) [4.64–6.08]	0.232	
Experience of success at work	6.3 ± 1.84 6.5 (1–9) [5.86–6.68]	5.9 ± 2.46 6 (1–9) [4.87–6.89]	4.5 ± 2.31 5 (1–9) [4.19–4.81]	<0.001	Practice owners/practice assistants < 0.001 Practice assistants/employed physicians = 0.008
Satisfaction with life	6.1 ± 1.98 6 (1–8) [5.64–6.54]	5.4 ± 2.02 5 (1–9) [4.53–6.19]	5.1 ± 2.02 5 (1–9) [4.82–5.36]	0.003	Practice owners/practice assistants < 0.001
Experience of social support	5.0 ± 2.06 5 (1–8) [4.51–5.44]	5.9 ± 2.14 6 (2–8) [5.04–6.80]	5.7 ± 2.01 6 (1–8) [5.38–5.92]	0.019	Practice owners/practice assistants = 0.010 Practice owners/employed physicians = 0.032

Notes. Grey marked fields mean values outside the normal range of the Stanine scores (4–6). Abbreviations: SD = standard derivation.

**Table 4 healthcare-12-00299-t004:** Gender distribution within AVEM patterns.

		Pattern G	Pattern S	Risk Pattern A	Risk Pattern B	Total
Male	n	5	17	4	4	30
	% from gender	16.7	56.7	13.3	13.3	100
	% from pattern	9.6	15.3	21.1	9.1	13.3
Female	n	47	94	15	40	196
	% from gender	24.0	48.0	7.7	20.4	100
	% from pattern	90.4	84.7	78.9	90.9	86.7

Notes. p_χ²_ = 0.517; (n = 226; pattern A n = 19, B n = 44, G n = 52, and S n = 111) with *p* = 0.445. Pattern combinations or subjects without assignments were not considered.

**Table 5 healthcare-12-00299-t005:** Spearman Rho correlations between AVEM dimensions and age, practice years, work experience, and hours per week.

AVEM Dimensions	Age	Practice Years	Work Experience	Hours per Week
Subjective importance of work	−0.102	−0.011	−0.104	0.124
Work-related ambition	−0.330 **	−0.299 **	−0.433 **	0.303 **
Willingness to work until exhausted	−0.076	−0.064	−0.146 *	0.264 **
Striving for perfection	−0.150 **	−0.175 **	−0.142 *	−0.009
Distancing ability	0.034	0.046	0.083	−0.337 **
Tendency to resign in the face of failure	−0.142 **	−0.193 **	−0.232 **	0.065
Proactive problem-solving	−0.130 *	−0.112 *	−0.148 *	0.010
Inner calm and balance	0.041	0.043	0.002	−0.010
Experience of success at work	0.103	−0.008	−0.090	0.011
Satisfaction with life	0.116 *	0.059	0.048	−0.305 **
Experience of social support	0.056	−0.050	0.146 *	−0.143 *

Notes. * The correlation is significant at the 0.05 level (two-sided). ** The correlation is significant at the 0.01 level (two-sided). Spearman’s correlation coefficients (rho): <0.39 low effect, 0.40–0.59 moderate effect, >0.6 means a strong effect. Red means negative correlations; green means positive correlations. The stronger the color, the stronger the correlation.

**Table 6 healthcare-12-00299-t006:** AVEM dimensions according to professional groups’ results—estimated marginal means and estimates of covariance parameters—of a GLM analysis.

AVEM Dimension	Practice Owner	Employed Physicians	Practice Assistants	Intraclass Correlation (Effect Size)	Sig.
	Mean ± SEM (95% Confidence Interval)				
Subjective importance of work	4.2 ± 0.24 [3.74–4.70]	3.9 ± 0.42 [3.11–4.74]	4.2 ± 0.16 [3.84–4.46]	0.074	0.117
Work-related ambition	5.0 ± 0.24 [4.49–5.44]	5.1 ± 0.41 [4.30–5.09]	4.8 ± 0.16 [4.46–5.09]	0.095	0.030
Willingness to work until exhausted	5.2 ± 0.26 [4.69–5.73]	4.8 ± 0.45 [3.90–5.68]	4.6 ± 0.16 [4.25–4.90]	0.051	0.251
Striving for perfection	3.6 ± 0.24 [3.12–4.08]	4.5 ± 0.41 [3.70–5.30]	4.8 ± 0.14 [4.53–5.06]	#	<0.001
Distancing ability	5.7 ± 0.24 [5.21–6.17]	6.2 ± 0.41 [5.41–7.01]	5.7 ± 0.17 [5.41–6.08]	0.142	0.010
Tendency to resign in the face of failure	4.2 ± 0.25 [3.71–4.71]	5.1 ± 0.43 [4.29–6.00]	4.6 ± 0.16 [4.28–4.89]	0.041	0.279
Proactive problem-solving	4.1 ± 0.24 [3.62–4.58]	4.0 ± 0.42 [3.17–4.84]	4.4 ± 0.14 [4.15–4.71]	0.002	0.953
Inner calm and balance	5.0 ± 0.23 [4.58–5.46]	5.5 ± 0.39 [4.80–6.33]	5.0 ± 0.14 [4.73–5.28]	0.050	0.218
Experience of success at work	6.2 ± 0.25 [5.74–6.73]	6.0 ± 0.44 [5.11–6.82]	4.7 ± 0.15 [4.42–4.99]	#	<0.001
Satisfaction with life	6.0 ± 0.23 [5.53–6.44]	5.5 ± 0.4 [4.75–6.32]	5.0 ± 0.14 [4.77–5.32]	0.029	0.397
Experience of social support	5.0 ± 0.23 [4.53–5.44]	5.9 ± 0.41 [5.10–6.71]	5.7 ± 0.14 [5.37–5.94]	0.013	0.728

Notes. SEM = standard error of the mean. # = this covariance parameter is unnecessary. Rather strong effect of occupational groups.

## Data Availability

There are no plans to grant access to full protocol, participant-level datasets, or statistical codes, as data contain potentially identifying information.

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
