# Peer review of "The Predominance of the Health-Promoting Patterns of Work Behavior and Experience in General Practice Teams—Results of the IMPROVEjob Study"

_healthcare, 2024, doi:10.3390/healthcare12030299_

Round 1

Reviewer 1 Report

Comments and Suggestions for Authors

Thank you for the opportunity to review your manuscript which is interesting and timely. Well written, researched and interesting. I have a couple of comments:

Abstract:

Remove all references not required in an abstract. 

Key words:

Remove Germany/AVEM/Survey/questionnaire.

Line 54 say for example (with reference to the studies mentioned in the following sentences)  

Line 121 say research undertaken 2017-2021 data collected 2017- ...data was not collected during COVID-19

Reviewer 2 Report

Comments and Suggestions for Authors

Thank you for allowing me to review this article of significant importance.

The article discusses an important topic in the field of health, which is the work related behavior and experience and its influence in health promotion.

I would like to offer some suggestions for improvement that I believe could benefit your paper.

·       Abstract

In the abstract, it would be important to identify the aim of the study and the period which the study was carried out. This information gives readers an overview of the study.

The text provided here follows the format of an academic paper, rather than that of a scientific journal abstract. To enhance its clarity, it would be best to eliminate numbers and references such as background, methods, results, and conclusions.

Not all the keywords are indexed terms. 

·       Introduction / Literature Review

The introduction is well structured, a literature review is carried out with some examples and rationale. The objective and contributions are clear. The definition of central concepts is short for their complexity, although are clear.

The justification for the study is clear and well-structured.

In line 112-114 it would be interesting to substantiate with some studies that practice owners may suffer additional stress when compared to the other members included in the study. It would be interesting to delve deeper into the differences between the different groups of participants in order to correctly interpret the data obtained.

·       Materials and Methods

The methodology chapter is well structured and the sub-chapters are clear.

It would be important to make explicit reference to ethical issues, such as the informed consent of the participants and guaranteeing the confidentiality and anonymity of the data.

In the generalized linear model, which variables were used and were any of them removed to obtain the best model? (line 209-210).

The study was developed with data from 60 general practices with 84 physicians in a leadership position, 28 employed physicians, and 254 practice assistants. How were these participants selected and what were the inclusion and exclusion criteria?

Results

Both the results and the discussion are extensive. However, they are well articulated and have adequate sub-chapters, which help to maintain the thread while reading the article.

AVEM's explanation is detailed and makes it possible to understand the dimensions assessed. However, it is only at the end of the article that the issues of validating the scale in the study population are mentioned.

Sometimes the graphics are not well explained in the text, please consider change the place of table 5 for instance, placing the interpretation text before the data table.

·       Discussion and conclusion

In the introduction you refer “We hypothesized that there would be significant differences between professional groups since the practice owners in particular bear greater and especially the economic responsibility, which can be associated with additional stress.” But in results and discussion you only refer that the “Practice affiliation had less than 5-10% effect on the results”. Was the hypotesis verified? If it hasn't been fully verified, how can we interpret this conclusion?

The comparison between your study and that of other professions is pleasant.

Acknowledging the study's limitations is relevant and is described clearly and transparently.

The recommendation to develop additional, more recent studies on the subject is very important, encouraging other researchers to continue studying the subject.

References

The cited references are recent, the majority of them, from the last 5 years, although some are from the last 10 years. The references previous to this time-space are from the framework and from the scales used in this study, giving substance.

I would like to congratulate the authors on this important paper.

Reviewer 3 Report

Comments and Suggestions for Authors

This is a really interesting paper and explores workplace experiences in greater detail than is possible with more widely used scales or measures of burnout and satisfaction. There was a lot to think about in the content and results, and thus the paper is very stimulating to those interested in improving the work experiences of health care professionals. Some of the differences between different work groups are a great entry point into talking about what different work stresses might affect people/groups differently.

A general comment is that sometimes the sentences are very long and not as clear as they could be in emphasizing the main point. This is just a general suggestion to be attentive to ways to make the authors' main points clearer or stronger if they have the opportunity to revise.

Another general comment: this study took place in the context of an intervention which strove to introduce both behavioral and structural changes. I was very interested to hear more about the intervention and would have liked to have some discussion of both 1) what structural changes accompanied the behavioral components (which are described in the discussion section) and 2) whether the authors think structural changes might effectively be tailored for structural interventions as well as behavioral ones - would they hypothesize that Risk Pattern A requires different work practice/work load changes than Risk Pattern B; furthermore, how can a practice accommodate both, given that both are present in the same practices?

Only a few specific comments:

- Table 3 is interesting and clear for a general readership, and it is appreciated that the immediately following table considers results by both job category and gender, as the intersection of role and gender was a question arising in my mind as I was reading. If possible, it might be worth discussing this intersection in the discussion - currently any discussion of this is absent.

- lines 414-420 make what seems to me to be an important point about the "prestige of primary care" but it is not clear what is meant here. Do providers in Germany feel there is not a lot of prestige in this area of medicine and that has a negative impact, or the converse? And what is the impact? It may not be possible to clarify this point, since it is a discussion of a different study.

- The finding that striving for perfection was higher among team associates than among physicians/practice owners is perhaps counter to what many would expect. This could also be an interesting point for more discussion.

- The discussion of the S pattern is generally balanced and informative in explaining that this pattern might call for intervention to improve motivation and thus work experience - but then again, it might have positive effects as well. I imagine it would take the authors far afield in their literature citations and theoretical considerations, but I would consider including a stronger acknowledgment/discussion that norms about how much one SHOULD be ambitious in work and find fulfilment in work vary both across and within cultures and social groups.

Comments on the Quality of English Language

The English is fine. Occasionally the wording is confusing or could be more succinct, but this is not a real concern. Only specific comment on language is that at line 467 the phrase "acting out" is used to describe an intervention and it is not clear what this means here.

Reviewer 4 Report

Comments and Suggestions for Authors

Comments 

1.     The sample size is modest (N=366).

2.     Please clarify the response rate of the survey in the revision. Non-response was random?

3.     Bias may arise from using self-reported data from a specific professional group.

4.     Discuss how the sample’s socio-demographic composition affects the generalizability of results.

5.     Empirical analysis does not establish causal relationships.

6.     Assess and discuss the plausibility of the correlation magnitudes.

7.     Non-random assignment of employees to workplaces/tasks most likely affects the results that are reported in the manuscript. This problem can be addressed by using information on employees’ wage and work histories (https://doi.org/10.1016/j.jebo.2012.09.005). This issue should be noted as a potential limitation of empirical approach.

8.     Address potential effect heterogeneity by age/gender, acknowledging the limitations due to sample size.

9.     Evaluate the results’ external validity for other high-income countries/contexts.
